# Boosting the Anticancer Activity of Sunitinib Malate in Breast Cancer through Lipid Polymer Hybrid Nanoparticles Approach

**DOI:** 10.3390/polym14122459

**Published:** 2022-06-16

**Authors:** Mohammed Muqtader Ahmed, Md. Khalid Anwer, Farhat Fatima, Mohammed F. Aldawsari, Ahmed Alalaiwe, Amer S. Alali, Abdulrahman I. Alharthi, Mohd Abul Kalam

**Affiliations:** 1Department of Pharmaceutics, College of Pharmacy, Prince Sattam Bin Abdulaziz University, P.O. Box 173, Al-Kharj 11942, Saudi Arabia; m.anwer@psau.edu.sa (M.K.A.); f.soherwardi@psau.edu.sa (F.F.); moh.aldawsari@psau.edu.sa (M.F.A.); a.alalaiwe@psau.edu.sa (A.A.); a.alali@psau.edu.sa (A.S.A.); 2Department of Chemistry, College of Science and Humanities, Prince Sattam Bin Abdulaziz University, P.O. Box 83, Al-Kharj 11942, Saudi Arabia; a.alharthi@psau.edu.sa; 3Nanobiotechnology Research Unit, College of Pharmacy, King Saud University, P.O. Box 2457, Riyadh 11451, Saudi Arabia; makalam@ksu.edu.sa; 4Department of Pharmaceutics, College of Pharmacy, King Saud University, Riyadh 11451, Saudi Arabia

**Keywords:** sunitinib, lipoid 90H, chitosan, nanoparticles, breast cancer, caspase

## Abstract

In the current study, lipid-polymer hybrid nanoparticles (LPHNPs) fabricated with lipoid-90H and chitosan, sunitinib malate (SM), an anticancer drug was loaded using lecithin as a stabilizer by employing emulsion solvent evaporation technique. Four formulations (SLPN1–SLPN4) were developed by varying the concentration of chitosan polymer. Based on particle characterization, SLPN4 was optimized with size (439 ± 5.8 nm), PDI (0.269), ZP (+34 ± 5.3 mV), and EE (83.03 ± 4.9%). Further, the optimized formulation was characterized by FTIR, DSC, XRD, SEM, and in vitro release studies. In-vitro release of the drug from SPN4 was found to be 84.11 ± 2.54% as compared with pure drug SM 24.13 ± 2.67%; in 48 h, release kinetics followed the Korsmeyer–Peppas model with Fickian release mechanism. The SLPN4 exhibited a potent cytotoxicity against MCF-7 breast cancer, as evident by caspase 3, 9, and p53 activities. According to the findings, SM-loaded LPHNPs might be a promising therapy option for breast cancer.

## 1. Introduction

Breast cancer is considered one of the leading types of cancer, surpassing lung cancer, as per worldwide cancer incidence in 2020 [1]. Breast cancer starts with the uncontrolled growth of breast cell in one or both sides. About one in eight women are diagnosed with breast cancer during their lifetime; the good news is that its curable if detected at an early stage [2]. Proliferation, apoptosis, angiogenesis, hypoxia, cancer stem cell activity, epithelial to mesenchymal transition (EMT), and metastasis are all related to the signaling system. Notch receptors and their ligands were shown to be overexpressed in breast cancer [3,4]. Signaling pathways upregulated leading to breast cancer include human epidermal growth factor receptor 2 (HER-2) tyrosine kinase pathway, a member of the ErbB family of transmembrane receptor tyrosine kinases [5]. The Hedgehog signaling pathway is also deregulated in breast cancer which is responsible for proper cell differentiation [6]. p53 mutation leads to aggressive diseases, such as breast cancer, playing a vital role in regulating the cell cycle, and apoptosis mutation of this gene causes cancer and shortens the overall survival. Another pathway actively involved in breast cancer is Phosphatase and tensin homolog (PTEN), reduced expression of which causes deceased formation of an enzyme phosphatase protein that acts as a tumor suppressor [7,8,9].

Sunitinib malate (SM) is a multiple tyrosine kinases inhibitor, used effectively in the cancer of the stomach, bowel, and esophagus, generally called gastrointestinal stromal tumors (GST), an abnormal proliferation of cells in gastrointestinal tract tissues [10,11]. It is a new vascular endothelial growth factor receptor practiced as a first-line therapy for advanced renal cell carcinoma [12] and a first-joint FDA-approved drug for these two indications. SM showed a promising activity against colorectal cancer, advanced non-small cell lung cancer (NSCLC) [13], hepatic cancer [14], and pancreatic neuroendocrine tumors (pNET) [15]. SM is one of the extensively studied antitumor agents in the breast cancer treatment with twenty-eight ongoing clinical trials, specifically sunitinib alone and in combinations. Sunhui et al.’s study showed sunitinib and curcumin have potential anticancer activity against breast cancer [16]. Sunitinib-loaded self-nanoemulsifying formulation has been developed with improved anticancer activity against the MCF-7 breast cancer cell line [17,18,19].

Moreover, breast cancer treatment failure could be surfaced due to poor drug solubility, low-bioavailability, permeability, cell uptake, drug resistance, and systemic toxicity. The cost of therapy and adverse drug effects could be lowered by adopting and aligning with new technologies. Nanotechnologies ensure it reduces adverse systemic toxic effects; thereby, the cost of therapy will also be reduced [20,21,22]. Nanocarriers enhance systemic drug circulation, and improve bioavailability, sustained release kinetics, and drug targeting at the receptor site. Enhanced permeability and retention effect facilitate the targeting of small molecules of nanosize and higher deposition of drug in cancer cells compared with the normal cells. Drug targeting involves the conjugate of chemotherapeutic-loaded nanocarrier with molecules that bind to the target (tumor) cell receptors [23,24,25,26]. Liposomes are thought to be a biocompatible vesicular structure with properties similar to biological membranes. Stability, low drug encapsulation, and burst drug release are the key concerns for vesicular systems. Polymer-based nanoparticles, on the other hand, are more stable than liposomes and also provide longer-lasting drug release. Synthetic (e.g., PLGA) and natural (e.g., chitosan) polymers are used to create polymeric nanoparticles. The preparation of lipid polymer hybrid nanoparticles (LPHNPs), which have both lipid and polymeric carriers, can address the constraints of both liposomes and polymeric nanoparticles. LPHNPs are next-generation core-shell nanostructures that are derived from both liposomes and polymeric nanoparticles (NPs), with a lipid coating encasing the polymer core loaded with drug that helps it to prolong systemic circulation and protect drug mitigation, and does not allow the water to obtain access into the drug-containing core. LPHNPs shows high entrapment, controlled release, cellular targeting, and serum stability. To the best of the author’s knowledge, no study has been conducted or reported on LPHNPs of SM [27,28,29]. Therefore, to facilitate targeting of SM at the tumor cell with higher drug loading, nanocarriers with a conjugate of lipid and polymer were selected in order to achieve higher antineoplastic activity with reduced toxicity to the normal cell at the target region. The objective of the current study was to prepare SM-loaded lipid–polymer hybrid nanoparticles using lipoid-90, chitosan, and evaluated for particle size, drug release, and anticancer activity against MCF7 cell lines.

## 2. Materials and Methods

### 2.1. Materials

Sunitininb malate (SM) was purchased from “Mesochem Technology” Beijing, China. Chitosan and polyvinyl alcohol (PVA) were procured from Sigma Aldrich, St. Louis, MO, USA. Lipoid 90H was a generous gift from Lyon, France. Human breast cancer cell line (MCF-7 cells) with estrogen, progesterone, and glucocorticoid receptors was procured from the American Type Culture Collection (ATCC). All other chemicals used were of analytical grade, and Milli-Q water was used wherever needed.

### 2.2. Preparation of SM-loaded Lipid–Polymer Hybrid Nanoparticles

Lipid–polymer hybrid nanoparticles (LPH-NPs) were prepared by emulsification solvent evaporation technique [30]. Briefly, pure SM (20 mg) and lipoid 90H (40 mg) was dissolved in 10 mL of dichloromethane to obtain the organic phase. Separately, the aqueous phase was prepared by dissolving chitosan (25–100 mg) and soyalecithin (20 mg) in 0.5% *w*/*v* acetic acid (Table 1). Further, prepared organic phase was emulsified into aqueous phase (with rate of 0.3 mL/min) using probe sonication (ultrasonic processor, Fisher scientific, Waltham, MA, United States) at power 65%, on–off cycle 5 sec, for 3 min. Formed emulsion was kept on a magnetic stirrer (500 rpm) at room temperature overnight. After complete evaporation of dichloromethane, reduced volume was centrifuged (HermleLabortechnik, Z216MK, Wehingen, Germany) at 6000 rpm for 15 h to obtain the sediment. Sediment pellet was then washed with milli-Q water thrice and lyophilized (Millirock Technology, Kingston, NY, USA) and lyophilized LPHNPs were collected for further analysis (Figure 1).

### 2.3. Measurement of Particle Size, Polydispersity Index (PDI) and Zeta Potential (ZP)

Freshly prepared SM-loaded LPHNPs were diluted 200 times with milli-Q water, sonicated for 5 min, and transferred into plastic cuvette, then analyzed for particle size and PDI using Malvern Zetasizer (ZEN-3600, Malvern Instruments Ltd., Worcestershire, UK). Three measurements were carried out for 11 runs with 10 sec durations each run at 25 °C temperature. The same procedure was followed for ZP measurements as particle size except a glass electrode sample holder was used instead of a plastic cuvette [31]. Each sample measurement was performed in triplicate.

### 2.4. Percent Drug Entrapment Efficiency (%EE)

Percent entrapment efficiency (%EE) was measured indirectly [20]. Freshly prepared SM-loaded LPHNPs dispersion were subjected to the centrifugation at 10,000 rpm for 15 min. Supernatant was then collected, pre-filtered by syringe filter 45 µm, then suitably diluted with methanol and quantified for unentrapped drug. Aliquots were analyzed using UV/Visible spectroscopy at λ_max_ 250 nm [31] (Jasco V630 UV/Visible spectrophotometer, Tokyo, Japan). The %EE was calculated using the following equations:(1)$$\%EE=\frac{Initial\;SM\;added\;in\;LPHNPs-Free\;SM\;in\;supernatant}{Initial\;SM\;added\;in\;LPHNPs}\times 100$$

### 2.5. Differential Scanning Calorimetry

Drug entrapment in the nanoparticles can be identified by differential scanning calorimetry (DSC). Samples 5 mg (pure SM, Lipoid 90H, chitosan, and optimized LPHNPs) (SLPN4) were packed in a hemispherical aluminium pan, separately. A pan filled with the sample was kept in a heating chamber against the blank. The temperature was raised from 50 °C to 350 °C at a rate of 20 °C/min; additionally, nitrogen gas was supplied at flow rate of 20 mL/min (Sinco 400, Seoul, Korea) [32]. Endothermic peaks were seen at the melting point of the sample; the temperature was then noted and studied.

### 2.6. FTIR Spectroscopy

FTIR spectra of SM, lipoid 90H, chitosan, and optimized LPHNPs (SLNP4) were taken using the KBr technique. The samples were mixed with crystalline KBr and the mixture was then compressed into transparent pellets using a hand-held compression machine. Thin transparent sample film enclosed in the die was fixed into the sample holder and scanned in the range of 400–4000 cm^−1^ (Jasco, V750, FTIR spectrophotometer, Tokyo, Japan). Peaks at the fingerprint region were interpreted, and additional or absent peaks were then studied for possible functional group interactions between drug and excipients. Spectrums were then collaged and presented for compatibility study [33].

### 2.7. XRD Diffraction Study

Sample (pure SM and SLPN4) were characterized by XRD to identify the nature of solid-state. X-ray diffractometer (Ultima IV, Rigaku Inc., Tokyo, Japan) was used with the following set parameters: Cu-kα radiation at 40 kV/40 mA, scan rate of 0.500°/min in the 0–60 (2θ) range, at a fixed monocromator (U4), attached with scintillation detector [33].

### 2.8. In-Vitro Release Studies

Comparative in vitro release studies of pure SM and optimized LPHNPs (SLPN4) were carried out using a dialysis bag (cut off mol wt. 12 kD) as per our previously reported method [30]. Briefly, pure SM and SLPN4 (equivalent to 20 mg of SM) were dissolved in pre-soaked dialysis containing 10 mL of phosphate buffer (pH 6.8), then dialysis bags were dipped into a beaker containing 40 mL of dissolution media and shaken on biological shaker (LBS-030S-Lab Tech, Kyonggi, Korea) at 100 rpm. At fixed time intervals (0.5, 1, 2, 3, 6, 12, 24, and 48 h), 1 mL of sample was withdrawn and compensated immediately with respective media to maintain sink condition, filtered, diluted, and analyzed at 250 nm [31]. Each sample was analyzed in triplicate. Further study of release mechanism was executed by fixing the release data into the following mathematical equations.
Qt = Q0 + k0t Zero order
logQt = logQ0 − kt/2.303 First order
Qt = kHt 1/2 Higuchi model
Mt/M∞ = ktn Korsmeyer Peppas model
where, Qt and Q0 represents (SM dissolved in media overtime t), (initial amount of SM dissolved in media, i.e., equal to zero). K marked constants of models. Mt and M∞ are cumulative drug release at time t and infinite time, respectively, t is the release time and n denotes the diffusional exponent indicating release mechanism [30].

### 2.9. Morphology

The morphology and size of images of optimized LPHNPs (SLPN4) were viewed by Scanning Electron Microscopic (SEM) (Zeiss EVO LS10; Cambridge, UK). In a thin film coater under vacuum, the sample was homogeneously dispersed and coated with gold-metal (Quorum Q150R S, Lewes, East Sussex, UK). The pre-treated sample was then bombarded with an electron beam, resulting in the creation of secondary electrons known as auger electrons. Only the electrons scattered at ≥90 degrees were picked from this interaction between the electron beam and the specimen’s atoms, and surface topography was obtained at 15 kV acceleration voltage and 7.58 K × magnification.

### 2.10. Cell Culture and Treatments

The Human breast cancer cell line (MCF7) was procured from the American Type Cutler Collection (Manassas, VA, USA). The cells were maintained in Dulbecco’s Modified Eagle Medium (DMEM) with phenol red supplement with 10% Fetal Bovine Serum (FBS), with Penicillin (100 units/mL), Streptomycin (100 μg/mL), and Amphotericin B (250 ng/mL), Gibco^®^ (New York, NY, USA). The cells were grown at 37 °C in 50 cm^2^ tissue culture flasks in a 5% CO_2_ humidified incubator. The cells were seeded into 96-well cell culture plates in DMEM.

### 2.11. MTT Assay on MCF7 Cells

To determine the dose dependent cell viability of MCF-7 cells, they were incubated with SM and SLPN4 ranging from 0.78 to 100 µg/mL (containing equivalent amount of SM drug) for 48h. The data presented demonstrate relative cell viability after the treatment, since MTT assay determines the viable cells through activity of mitochondria. This is primarily targeted through mitochondria-mediated apoptosis. Thus, this approach was adopted to determine the activity of SM and SLPN4. The IC_50_ values were calculated using Log (inhibitor) versus normalized response on variable slope by GraphPad Prism V-5.1 (San Diego, CA, USA).

### 2.12. Morphological Changes on MCF-7 Cells

The cytotoxic effect of pure STB and SLPN4 was also determined by visualizing the morphological changes in MCF7 cells [34]. The IC_50_ value of SM (10.79 µg/mL) equivalent to SLPN4 (8.24 µg/mL) was taken as the dose of treatment and morphological features were manifested by phase-contrast microscopy. Morphological features such as membrane blebbing, cell shrinkage, and necrosis were determined.

### 2.13. Caspase-3, Caspase-9 and p53 Assay by ELISA

Caspase-3, caspase-9, and p53 assay ELISA kits were used to measure caspase activity [35]. The MCF-7 cells (50,000 cell/well) were seeded in 96-well plates. The cells were cultured for 24 h at 37 °C in a humidified incubator with 5% CO_2_. The SM, SLPN4-treated, and untreated control cells were then allowed to equilibrate at room temperature in 96-well plates. Each well of plate (SM, SLPN4-treated, and control) containing 100 µL of culture media received 100 µL of caspase-3 and 9 reagents. The plate was covered and the contents were stirred at 500 rpm for 30 s. After 30 min of incubation at room temperature, the optical density was measured at 405 nm using an an ELISA reader.

### 2.14. Stability Study

To analyze the change in formulation over storage or shelf life, a stability study was conducted for optimized LPHNPs (SLPN4). The formulation was sealed in a glass vial and stored for three months at 25 ± 0.5 °C/65 ± 5% RH and 40 ± 2 °C/75 ± 5% RH in a stability chamber, and physical appearance, particle size, PDI, ZP and entrapment were examined in samples obtained at 0, 1, 2, and 3 months [36,37].

### 2.15. Statistical Analysis

The experimental data were analyzed statistically using one way ANOVA followed by Tukey’s multiple comparison test using SPSS 16 software (version 25.0; SPSS Inc., Chicago, IL, USA) (*p* < 0.01) was considered significant.

## 3. Results and Discussion

### 3.1. Measurement of Particle Size, Polydispersity Index (PDI) and Zeta Potential (ZP)

SM-loaded LPHNPs (SLPN1-SLPN4) were prepared by the single emulsification method. The developed LPHNPs were characterized for their particle size, PDI, and ZP and measured in the range of 218–439 nm, 0.269–0.504, and +18 to +34 mV, respectively (Figure 2). According to studies, nanoparticles with a size range of 40 to 400 nm are appropriate for ensuring long circulation duration and increased accumulation of drug in tumors with limited renal clearance [38,39]. The positive values of ZP was measured due to the amino group present on the surface of chitosan polymer [40,41]. From the results, it was observed that increase in concentration of chitosan in formulations increased the size of particles.

### 3.2. Percent Drug Entrapment Efficiency (%EE)

The %EE of SM-loaded LPHNPs (SLPN1–SLPN4) were measured in the range of 45.71 ± 3.3–83.03 ± 4.9% (Figure 2). The highest drug entrapment (83.03 ± 4.9%) was measured in SLPN4, the large amount of chitosan (100 mg) used in this formulae expected to prevent leakage of drug from polymeric core, thereby improving the entrapment efficiency of drug [42]. Among the developed LPHNPs, SLPN4 was optimized with size (439 ± 5.8 nm), PDI (0.269), ZP (+34 ± 5.3 mV), and EE (83.03 ± 4.9%) and further evaluated.

### 3.3. Differential Scanning Calorimetry

DSC spectra of pure SM, Lipoid 90H, chitosan, and optimized LPHNPs (SLPN4) are presented in Figure 3. DSC studies confirmed the crystallinity and amorphocity nature of the sample, which indicates the encapsulation of drugs in nanoparticles [20,43]. Pure drug SM exhibited a sharp endothermic peak at 205 °C, which indicates its melting temperature [44,45]. The SM peak completely disappeared in DSC spectra of SLPN4, confirming SM entrapment in LPHNPs. Sharp endothermic and broad exothermic peaks of lipoid 90H and chitosan could be seen near 193 °C and 320 °C, respectively.

### 3.4. FTIR Spectroscopy

Figure 4 indicates the FTIR spectra of pure SM, lipoid 90H, chitosan, and optimized LPHNPs (SLPN4) in the wavelength range of 400–4000 cm^−1^. The FTIR spectra of pure SM showed many intense peaks at 3324 cm^−1^ for the acidic O-H, 2981 cm^−1^ for the acidic –CH=CH- (aryl) str, 2884 cm^−1^ for C-H (alkyl) str, 1635 cm^−1^ for the –NHCO str, and 1073 cm^−1^ bands correspond to the (C–F stretching) [41]. FTIR spectra of chitosan showed a broad peak at 3579 cm^−1^ (-OH str), 2873 cm^−1^ (CH_2_ str) [46]. The FTIR spectra of phospholipon 90H exhibited characteristic peaks at 2934 cm^−1^ and 2861 cm^−1^ for –C-H- str present in long fatty acid chain, 1722 cm^−1^ for –C=O str, and 972 cm^−1^ for P=O str [47]. In the optimized LPHNPs (SLP4), peaks at 3324 cm^−1^, 2981 cm^−1^, and 2884 cm^−1^ were found to have disappeared. The FTIR peaks of SLPN4 represents no significant modifications in the functional peaks of the SM in the lipid–polymeric NPs matrix, thereby retaining drug’s physicochemical properties and efficient chemical stability for the encapsulated SM in the fabricated nanocarriers. As there is no evidence of drug–polymers interaction, the selected lipid, polymer, and drug have the compatibility and suitability for the SM-loaded HNPs.

### 3.5. XRD Diffraction Study

Pure drug sunitinib (SM) showed intense X-ray diffractions at 13.3°, 19.5°, 22.4°, and 25.6° at 2θ, which represents the crystalline state of the drug [48]. However, the optimized LPHNPs (SLPN4) diffractogram also showed few peaks with reduced intensities, which are attributed to the amorphous state of drug in the nanoparticles due to the destruction of crystalline nature of SM and molecular dispersion of polymers, lipid, and drug (Figure 5).

### 3.6. In-Vitro Release Studies

In vitro drug release studies were conducted at pH 6.8, i.e., the pH of cancer cells [49]. A rapid drug release (98.45%) was exhibited by pure SM at the end of 6 h as compared with SLPN4. The optimized formulation (SLPN4) showed an initial rapid release of the drug for the first 6 h of study, followed by a sustained drug release. Initial rapid release could be due to the adsorbed drug over the polymer, which dissociated easily in the diffusion medium and released [20] (Figure 6).

Release kinetics were assessed for optimized LPHNPs (SLPN4). The goodness of fit models was selected by evaluating R^2^ value in the prediction of the release mechanism. The kinetics analysis of regression coefficient of all the four models used indicated R^2^ values for zero order (0.612), first order (0.9359), Higuchi model (0.8494), and Korsmeyer–Peppas (0.9406) with diffusion coefficient n (0.271). The optimized LPHNPs (SLPN4) followed the Fickian diffusion (n < 0.5) and mechanism of release from the Korsmeyer–Peppas kinetic model [30].

### 3.7. Morphology

SEM images of optimized LPHNPs (SLPN4) are shown in the Figure 7. It was confirmed that optimized LPHNPs (SLNP4) were small and spherical in shape with aggregation, probably due to the presence of lipids. The size observed by SEM was approximately same as that measured by the DLS method.

### 3.8. MTT Assay on MCF7 Cells

The MTT assay showed concentration-dependent reduction in cell viability for SM and optimized SM-loaded LPHNPs (SLPN4) against MCF-7 cell lines (Table 2 and Figure 8). The IC_50_ values for pure drug SM and SLP4 were found as 10.79 and 8.24 µg/mL for MCF-7 cells, respectively. The formulation SLP4 showed a significant reduction in cell viability (80.52%, 73.58%, 65.89%, 61.60%, 47.53%, 24.97%, 13.63%, and 6.79% at 0.8, 1.6, 3.1, 6.3, 12.5, 25, 50, and 100 µg/mL) in comparison with pure drug SM (92.05%, 85.26%, 75.35%, 67.49%, 54.04%, 32.94%, 15.24%, and 8.17% at 0.8, 1.6, 3.1, 6.3, 12.5, 25, 50, and 100 µg/mL), respectively, against MCF-7 cells. Based on the results of MTT assay, it was observed that SLP4 exhibited potential anticancer activity against breast cancer cell lines, probably due to the enhancement of the release of SM from the SLP4 formulation. SM-loaded LPHNPs (SLPN4) could be used as a potent carrier for the treatment of breast cancer.

### 3.9. Morphological Changes on MCF-7 Cells

The morphological changes on MCF-7 cell lines after treatment of control, pure SM, and SLPN4 are presented in Figure 9, and after 24 h of incubation, SLPN4-treated cells evidenced maximum cell death in comparison with pure SM and control. The morphological changes observed in MCF-7 by SLPN4 were due to damage in cell organelles. The morphological changes were observed in MCF7 cells by SM as it is a tyrosine kinase inhibitor and has an anti-proliferative effect on breast cancer cell lines (MCF-7). The SM and SLPN4 treatments cause concentration-dependent cell growth suppression due to apoptosis, as made evident by Caspase-3, p53, and Caspase-9 levels in MCF7. These results are consistent and similar in previous reports [50,51].

### 3.10. Caspase-3, Caspase-9 and p53 Assay by ELISA

Apoptosis is a type of programmed cell death that involves the disassembly of intracellular components while avoiding injury and inflammation to nearby cells [52,53]. The activation of caspase-3, -9, and p53 are mainly responsible for cancer cell apoptosis [54,55]. In this study, an increase in caspase 3, 9, and p53 production in pure SM- and SLPN4-treated MCF-7 cells was compared to the untreated control group to confirm apoptosis. When MCF-7 cells were exposed to pure SM and SLPN4, their caspase 3, 9, and p53 activity was found four, fourteen, and seven times higher compared with control (untreated) cells, respectively (Figure 10). Enhanced effectiveness of SM in LPHNPs suggests a possible reason for induction of apoptosis in cancer cells.

### 3.11. Stability Study

Following ICH guidelines, the stability of optimized LPHNPs (SLPN4) was assessed for three months in terms of particle size, PDI, ZP, and entrapment efficiency after storage. Table 3 shows the stability study parameters for LPHNPs (SLPN4), which were all within acceptable ranges, indicating that the developed formulation was stable for three months. A significant change (** *p* < 0.01) in particle size and PDI were noted when stored at 25 ± 0.5 °C/65 ± 5% RH and 40 ± 2 °C/75 ± 5% RH, which indicated that high temperature can lead to aggregation of the nanoparticles.

## 4. Conclusions

No research has been reported on SM as a breast cancer treatment using lipid–polymer nanoparticles. In vitro release and cell line investigations showed that the developed optimized SM-loaded LPHNPs (SLPN4) significantly enhanced the release and accessibility of SM at the breast cancer cells. Importantly, the potential cytotoxicity of SLPN4 on the MCF-7 breast cancer cell line was determined using the MTT test for cytotoxicity, ELISA activity of caspase-3, -9, and p53 in comparison to free SM or control. Thus, the developed formulation could provide an attractive nanoplatform for the treatment of breast cancer and may be the focus for the future chemotherapeutic investigations.

## Figures and Tables

**Figure 1 polymers-14-02459-f001:**
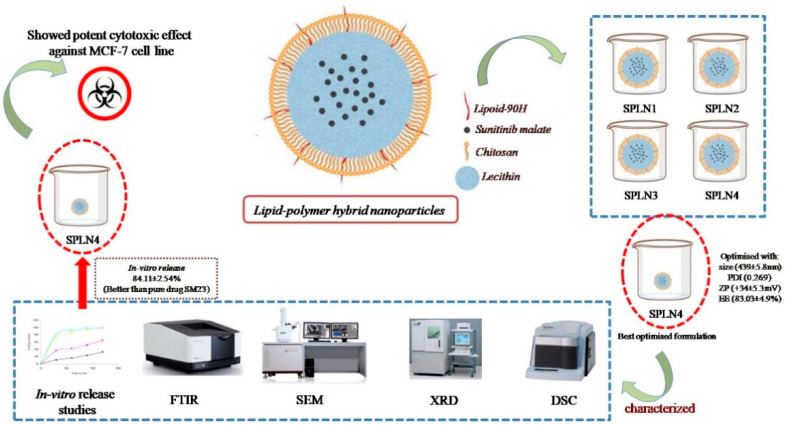
Schematic diagram of SLPNs synthesis.

**Figure 2 polymers-14-02459-f002:**
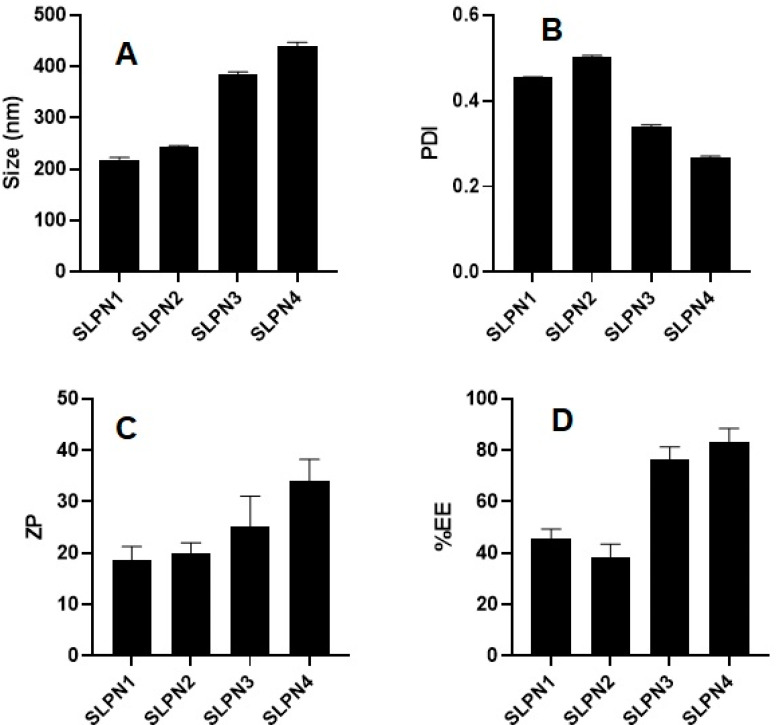
Particle size, PDI, ZP, and %EE of developed SM-loaded lipid polymer nanoparticles (SLPN1–SLPN4) tested with one way ANOVA followed by Tukey’s multiple comparison between formulations. (**A**) Particle size—significant difference (** *p* < 0.01) among formulations. (**B**) PDI—significant difference (** *p* < 0.001) among formulations. (**C**) Zeta potential—results are not significant among (SLPN1 vs. SLPN2, SLPN1 vs. SLPN3, SLPN2 vs. SLPN3, and SLPN3 vs. SLPN4 formulations) and significant (** *p* < 0.01) between SLPN1 vs. SLPN4 and SLPN2 vs. SLPN4. (**D**) %EE—results are not significant among (SLPN1 vs. SLPN2 and SLPN3 vs. SLPN4 formulations) and the rest are significant (** *p* < 0.01).

**Figure 3 polymers-14-02459-f003:**
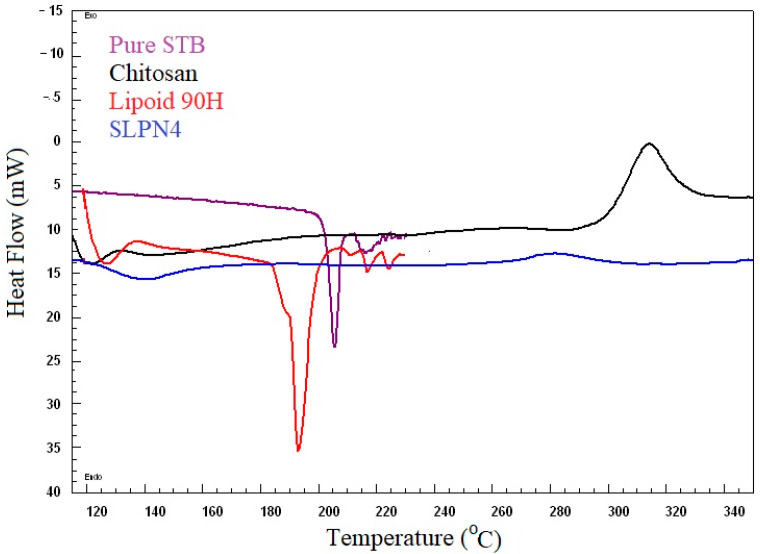
Comparative DSC spectra of pure SM, SLPN4, lipoid 90H, and chitosan.

**Figure 4 polymers-14-02459-f004:**
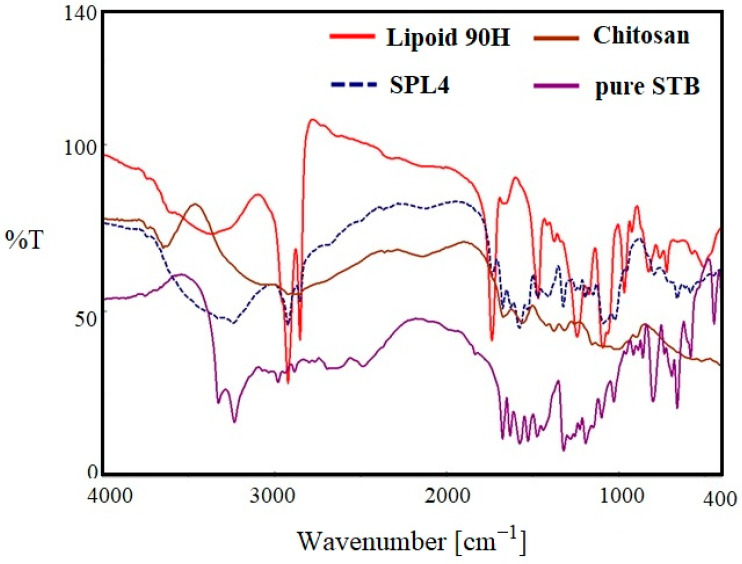
Comparative FTIR spectra of pure SM, SLPN4, lipoid 90H, and chitosan.

**Figure 5 polymers-14-02459-f005:**
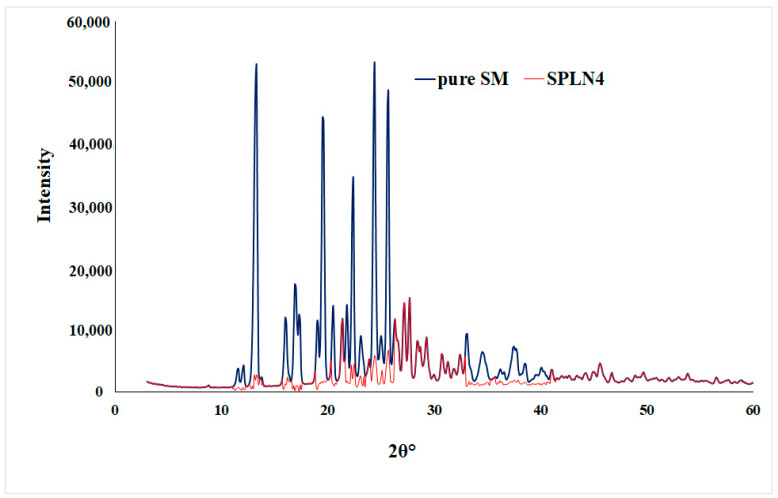
XRD spectra of pure SM and optimized LPHNPs (SLPN4).

**Figure 6 polymers-14-02459-f006:**
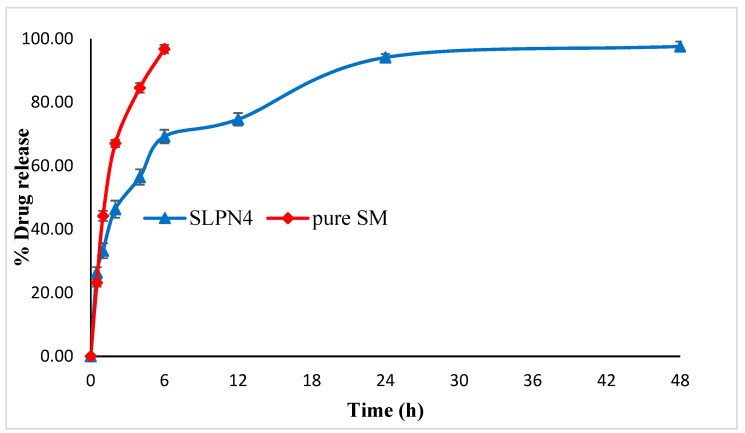
Comparative in vitro release profile of pure SM and optimized LPHNPs (SLPN4).

**Figure 7 polymers-14-02459-f007:**
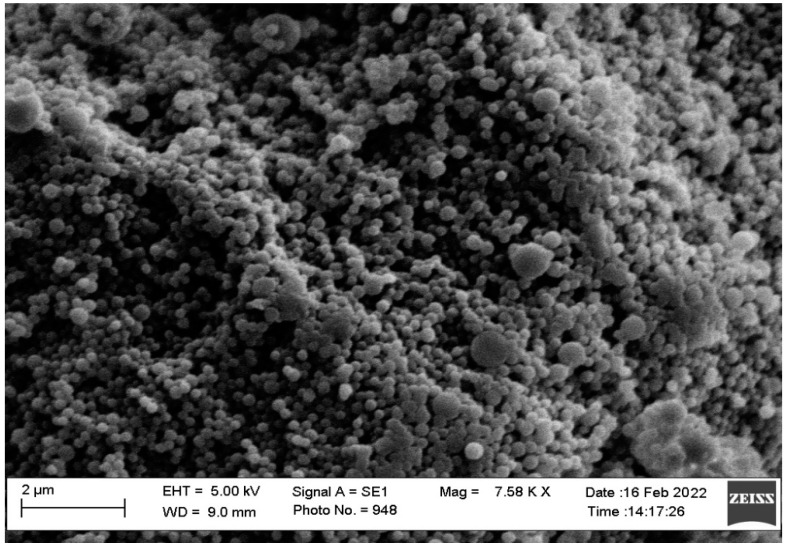
SEM images of optimized LPHNPs (SLPN4).

**Figure 8 polymers-14-02459-f008:**
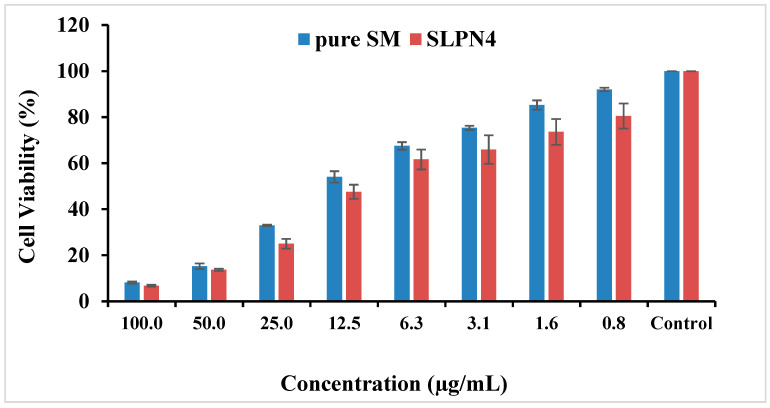
Cytotoxicity of pure SM and SLPN4 after 48 h incubation with MCF-7 cells at concentrations 0.8–100 µg/mL.

**Figure 9 polymers-14-02459-f009:**
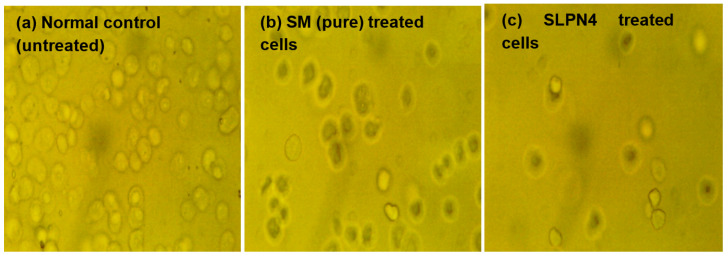
Morphological changes in MCF7 cells after exposure to Sunitinib-containing formulations. Untreated cells/control, cells incubated with pure SM, and SLPN4.

**Figure 10 polymers-14-02459-f010:**
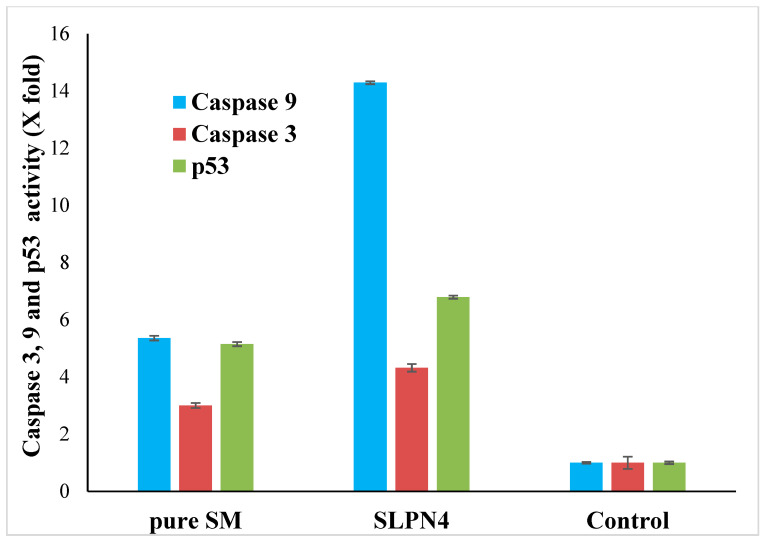
Activation of caspase 3, 9, and p53 in pure SM- and SLPN4-treated MCF7 cells was compared to the untreated control. SLPN4 (*p* < 0.05) vs. free SM and control group.

**Table 1 polymers-14-02459-t001:** Composition of SM-loaded LPHNPs.

Composition (mg)	LPHNPs			
	SLPN1	SLPN2	SLPN3	SLPN4
Sunitinib	20	20	20	20
Lipoid 90H	40	40	40	40
Soyalecithin	20	20	20	20
Chitosan	25	50	75	100

**Table 2 polymers-14-02459-t002:** Percent cell viability against concentration.

Conc (µg/mL)	% Cell Viability	
	Pure SM	SLPN4
100.0	8.176 ± 0.457	6.793 ± 0.392
50.0	15.242 ± 1.188	13.636 ± 0.478
25.0	32.944 ± 0.305	24.976 ± 2.106
12.5	54.046 ± 2.463	47.537 ± 3.091
6.3	67.492 ± 1.627	61.607 ± 4.334
3.1	75.356 ± 0.885	65.893 ± 6.200
1.6	85.265 ± 2.041	73.585 ± 5.619
0.8	92.055 ± 0.763	80.511 ± 5.431
Control	100.000 ± 0.000	100.000 ± 0.000

**Table 3 polymers-14-02459-t003:** Stability data of optimized LPHNPs (SLPN4).

Months	Storage Conditions	Particle Size	PDI	ZP (mV)	%EE
0	-	439 ± 5.8	0.269 ± 0.00052	+34 ± 5.3	83.03 ± 4.9
1	25 ± 0.5 °C/65 ± 5% RH	468 ± 4.5 **	0.267 ± 0.00052 ^ns^	+33 ± 4.1 ^ns^	82.42 ± 4.9 ^ns^
2		472 ± 7.4 **	0.263 ± 0.00179 **	+31 ± 3.2 ^ns^	78.11 ± 7.4 ^ns^
3		476 ± 8.5 **	0.278 ± 0.00089 **	+29 ± 2.8 ^ns^	77.28 ± 4.8 ^ns^
1	40 ± 2 °C/75 ± 5% RH	467 ± 2.5 **	0.289 ± 0.00288 *	+31 ± 6.1 ^ns^	81.45 ± 3.3 ^ns^
2		473 ± 6.8 **	0.312 ± 0.00137 **	+24 ± 5.4 ^ns^	74.32 ± 3.1 ^ns^
3		476 ± 7.5 **	0.334 ± 0.00358 **	+21 ± 4.3 ^ns^	71.31 ± 5.9 ^ns^

Significant difference (** *p* < 0.01) compared with month 0; non-significant (^ns^) compared with month 0.

## Data Availability

The data presented in this study are available on request from the corresponding author.

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
