# Peer review of "Boosting the Anticancer Activity of Sunitinib Malate in Breast Cancer through Lipid Polymer Hybrid Nanoparticles Approach"

_polymers, 2022, doi:10.3390/polym14122459_

Round 1
Reviewer 1 Report
Summary
I appreciate the research team for this interesting work in the therapeutic field, particularly in cancer, which is the leading cause of death worldwide. Breast cancer is found to be the second most common cancer in women after skin cancer. Breast cancer became the most common cancer globally as of 2021, accounting for 12% of all new annual cancer cases worldwide, according to the World Health Organization.
The work presents the development of lipid polymer hybrid nanoparticles (LPHNPs) to improve antineoplastic activity. The emulsification- solvent evaporation method chosen for the fabrication of nanoparticles is simple and flexible. The results of newly developed LPHNPs have shown improvements in the drug release profiles compared to the pure sunitinib malate (SM). All the LPHNPs are spherically shaped, and the size was confirmed by both SEM and DLS methods. The fabricated nanoparticles are well characterized by SEM, FTIR, XRD, and DSC. Among the four formulations, SPLN4 has been reported to possess excellent results with the drug entrapment efficiency (83.03+4.9%), size (439+5.8), PDI (0.269), and ZP (+34+5.3). The kinetics studies of drug release follow the Korsmeyer-Peppas model with the Fickian release mechanism. An improved cumulative drug release of 84.11+2.38% was reported for SLPN4 compared to pure SM (24.13+2.31%) at 48hr with burst release of the first 30 minutes. MTT assay on MCF7 cells has shown higher anti-cancer activity for SLPN4 than pure SM. Caspase-3, caspase-9, and p53 assay by ELISA kit have shown an increased production of caspase-3, caspase-9, and p53 in cells treated with SLPN4 compared to untreated cells, thus enhancing the apoptosis in tumor cells. The production of these three proteins in cells by SPLN4 was in the order caspase-9>p53>caspase-3. The stability study states that formulation at 25+0.5°C/65+5%RH storage conditions is more stable than formulation at 40°C/75+5% RH with a smaller decrease in the size, PDI, ZP, and %EE. The experiment with varying concentrations of chitosan polymer (four formulations) proves that the size of the nanoparticle and the percent of drug entrapment efficiency is directly proportional to the chitosan concentration. A good SEM image of SLPNs is also provided.
The overall structure of the manuscript is good. The abstract itself gives a short description of the research work. The research methodology and the corresponding results are well presented. The references cited are appropriate.
Thus, the work imparts the possible applications of chemotherapeutic drugs known for a particular type of cancer to other cancer types also. Further modifications to the core or shell of nanoparticles can provide the incorporation of multiple drugs into the nanostructures. Hence, this work opens a new path to improving therapeutic investigations in combination with nanotechnology.
Issues
- According to lines 260-261, broad endothermic and exothermic peaks for lipoid 90H and chitosan near 138°C and 282°C respectively should be in the DSC spectra (line 262), presented in figure.1. But DSC spectra show no peaks near 138°C and 282°C for lipoid 90H and chitosan. But the spectra show endothermic and exothermic peaks for SLPN4 near those temperatures. Please see the attached file and correct.
- In the data regarding the reduction in cell viability, the percentage of reduction in cell viability corresponding to the 6.3µg/ml for optimized SLPNs is not provided. Please check carefully the correspondence between concentrations and cell viability data. Please check the table provided in the attached file.
- In line 277, the correction of sentence starts with AS there is no evident…… into As there is no evident…….
- In line 134, the spelling of nanoparticles.
- Line 305 and 307 “korsmeyer peppas” please correct in “Korsmeyer Peppas” and “Peppas” and please add a reference.
- Please correct Fig.4 according the above mentioned models. This graphs show a “zero order kinetics” but in the discussion it seems to be different. Please check and correct.
Queries
- The developed nanoparticles are lipid-polymer nano micelles with a polymer core and lipid shell.
- Are there any pH conditions for the preparation of SM-LPHNPs?
- What about the critical micellar concentration of micelles?
- The size of the nanoparticle is the critical element in the tumor penetration. The studies show that NPs with too small particle sizes will be cleared by RES and NPs with sizes greater than 150nm are not internalized into the cell. The optimized SLPN4 has a size of 439+8 nm.
- What about the cell uptake of the developed LPHNPs?
- Explanation for degradation mechanism of LPHNPs.
- Does the developed LPHNPs possess any stimuli-responsive property.
Suggestions
- The sentence of line 44-48 is too long. Please separate these lines into 2 sentences to emphasize the significance of the sunitinib malate drug.
- Addition of image of sunitinib malate and schematic representation of the synthesis of LPHNPs.
- Tabulation of cell viability data of pure SM and SLPN4.
- Need more details about breast cancer and signalling pathways in the introduction part.

Author Response
Reviewer 1
Dear Sir/madam,
We appreciate the reviewer for your precious time in reviewing our paper and providing valuable comments. It was your valuable and insightful comments that led to possible improvements in the current version. The authors have carefully considered the comments and tried our best to address every one of them.
We hope the manuscript after careful revisions meet your high standards. The authors welcome further constructive comments if any. Below we provide the point-by-point responses. All modifications in the manuscript have been done in track change.
Issues
- According to lines 260-261, broad endothermic and exothermic peaks for lipoid 90H and chitosan near 138°C and 282°C respectively should be in the DSC spectra (line 262), presented in figure.1. But DSC spectra show no peaks near 138°C and 282°C for lipoid 90H and chitosan. But the spectra show endothermic and exothermic peaks for SLPN4 near those temperatures. Please see the attached file and correct.
Answer: Thanks for pointing this out. A correction has been made in revised manuscript. “A sharp endothermic and broad exothermic peaks of lipoid 90H and chitosan could be seen near 193 °C and 320 °C, respectively”.
2. In the data regarding the reduction in cell viability, the percentage of reduction in cell viability corresponding to the 6.3µg/ml for optimized SLPNs is not provided. Please check carefully the correspondence between concentrations and cell viability data. Please check the table provided in the attached file.
Answer: Cell viability missing data of SLPN4 has been corrected at concentration of 3.1 µg/ml (80.52, 73.58, 65.89, 61.60, 47.53, 24.97, 13.63 and 6.79% at 0.8, 1.6, 3.1, 6.3, 12.5, 25, 50 and 100 µg/mL).
3. In line 277, the correction of sentence starts with AS there is no evident…… into As there is no evident…….
Answer: Corrected.
4. In line 134, the spelling of nanoparticles.
Answer: Corrected.
5. Line 305 and 307 “korsmeyer peppas” please correct in “Korsmeyer Peppas” and “Peppas” and please add a reference.
Answer: Corrected and suitable reference has been added.
6. Please correct Fig.4 according the above mentioned models. This graphs show a “zero order kinetics” but in the discussion it seems to be different. Please check and correct.
Answer: We repeated the drug release study as suggested by Editor. The kinetics analysis of regression coefficient of all the four models used indicated R2 value for zero order (0.612), first order (0.9359), Higuchi model (0.8494), Korsmeyer Peppas (0.9406) with diffusion coefficient n (0.271). The optimized LPHNPs (SLPN4) followed the Fickian diffusion (n<0.5) and mechanism of release from Korsmeyer Pep-pas kinetic mode
Queries
- The developed nanoparticles are lipid-polymer nano micelles with a polymer core and lipid shell.
- Are there any pH conditions for the preparation of SM-LPHNPs?
Answer: Not any pH condition studied during preparation.
- What about the critical micellar concentration of micelles?
Answer: CMC of micelle not measured.
2. The size of the nanoparticle is the critical element in the tumor penetration. The studies show that NPs with too small particle sizes will be cleared by RES and NPs with sizes greater than 150 nm are not internalized into the cell. The optimized SLPN4 has a size of 439+8 nm.
Answer: To passively deliver nanoparticles to tumor sites, the enhanced permeability and penetration (EPR) effect is the strategy that is mostly used, which is specific only in tumors due to the rapid proliferation of tumor cells and the abnormal tumor vasculature system. Nanoparticles with large sizes tend to be more capable of retention in tumor tissue than those with smaller sizes. [Perrault S. D.; Walkey C.; Jennings T.; Fischer H. C.; Chan W. C. Mediating tumor targeting efficiency of nanoparticles through design. Nano Lett. 2009, 9 (5), 1909–1915. 10.1021/nl900031y].
- What about the cell uptake of the developed LPHNPs?
Answer: We appreciate the suggestion of the reviewer.In future studies of hybrid NPs of Sunitinib, we will perform cell uptake.
3. Explanation for degradation mechanism of LPHNPs.
Answer: The polymer coating over lipid particles has enormous benefits that include the protection, promotion of durability, reduction in drug degradation due to leakage, and sustained the drug release.
4. Does the developed LPHNPs possess any stimuli-responsive property?
Answer: Yes, LPHNPs possess endogenous (pH, redox, hypoxia, and reactive oxygen species) and exogenous (light, temperature, magnetic field, and ultrasound) stimuli in drug delivery applications. It is thoroughly discussed by “ Ayeskanta Mohanty, Saji Uthaman, In-Kyu Park, Chapter 12 - Lipid–polymer hybrid nanoparticles as a smart drug delivery platform, Editor(s): Virendra Gajbhiye, Kavita R. Gajbhiye, Seungpyo Hong, Stimuli-Responsive Nanocarriers, Academic Press, 2022, Pages 319-349”.
Suggestions
- The sentence of line 44-48 is too long. Please separate these lines into 2 sentences to emphasize the significance of the sunitinib malate drug.
Answer: Done.
2. Addition of image of sunitinib malate and schematic representation of the synthesis of LPHNPs.
Answer: As suggested, a schematic representation of the synthesis of LPHNPs as Figure 1 has been added in revised manuscript.
3. Tabulation of cell viability data of pure SM and SLPN4.
Answer: As suggested a table of cell viability has been inserted in revised manuscript.
Table 2: Percent cell viability Vs concentration
Conc (µg/mL) |
% Cell Viability |
|
Pure SM |
SLPN4 |
|
100.0 |
8.176±0.457 |
6.793±0.392 |
50.0 |
15.242±1.188 |
13.636±0.478 |
25.0 |
32.944±0.305 |
24.976±2.106 |
12.5 |
54.046±2.463 |
47.537±3.091 |
6.3 |
67.492±1.627 |
61.607±4.334 |
3.1 |
75.356±0.885 |
65.893±6.200 |
1.6 |
85.265±2.041 |
73.585±5.619 |
0.8 |
92.055±0.763 |
80.511±5.431 |
Control |
100.000±0.000 |
100.000±0.000 |
4. Need more details about breast cancer and signalling pathways in the introduction part.
Answer: Added in revised manuscript.

Reviewer 2 Report
In this manuscript, authors performed Boosting the Anticancer Activity of Sunitinib malate in Breast Cancer through Lipid Polymer Hybrid Nanoparticles Approach. In my opinion, some issues should be further address and I hope following comments could be helpful for improving their paper.
- It would be better if author add graphical representation of overall study as schematic digram to draw readers attention.
- In introduction section Information about cancer and Anti cancer drug delivery is little, kindly cite some recent litreature. https://pubs.rsc.org/en/content/articlelanding/2019/bm/c9bm00139e/unauth, https://pubs.rsc.org/en/content/articlelanding/2019/tb/c9tb01842e/unauth
- Why drug released was performed only in pH 6.8?
- In materials sections authors need to add Cell section, properly added all cell used in this manuscript along with cell culture protocols.
- Why morphological changes observed in breast cancer?
- Authors also need to perform in vitro cytotoxicity study of hybrid Nps on Normal cell to confirm safety.
- for size, PDI, and ZP results authors need to draw proper graphs on graphpad. Kindly remove Table 2. for these results authors also need to insert proper results obtained from the zeta sizer and DLS.
- For drug release study how the sample were analyzed by using 250 nm? Which instrument is used, HPLC or UV?
- Methodology for In vitro drug release study need to re write again, kindly add time interval b/w two samples taken and after taken sample how to maintain sink condition?
- Discusion for SEM result need improvement.
- In fig 7 there is no morphological diffrence observed, kindly performed it again.
- Its better draw all graphs by using Graphpad and apply all the T test, Excel graphs are not attractive.
- Why not author checked this Hybrid Nps loaded with anti cancer drug on Animal? Molecular studies such as H&E staining is missing. Kindly perform it,
- Discussion part need to revised to and compare your results with already published literature to support your hypothesis.
- Please revisit the entire manuscript for minor grammar and typo issues.
Author Response
Dear Sir/madam,
We appreciate the reviewer for your precious time in reviewing our paper and providing valuable comments. It was your valuable and insightful comments that led to possible improvements in the current version. The authors have carefully considered the comments and tried our best to address every one of them.
We hope the manuscript after careful revisions meet your high standards. The authors welcome further constructive comments if any. Below we provide the point-by-point responses. All modifications in the manuscript have been done in track change.
- It would be better if author add graphical representation of overall study as schematic diagram to draw reader’s attention.
Answer: As suggested, a schematic representation of the synthesis of LPHNPs as Figure 1 has been added in revised manuscript.
2. In introduction section Information about cancer and Anti cancer drug delivery is little, kindly cite some recent litreature. https://pubs.rsc.org/en/content/articlelanding/2019/bm/c9bm00139e/unauth, https://pubs.rsc.org/en/content/articlelanding/2019/tb/c9tb01842e/unauth
Answer: Suggested references has been added in introduction of revised manuscript.
3. Why drug released was performed only in pH 6.8?
Answer: pH 6.8 is favorable condition to target cancerous cells as discussed by Arora et al. [Arora, S.; Saharan, R.; Kaur, H.; Kaur, I.; Bubber, P.; Bharadwaj, L.M. Attachment of Docetaxel to Multiwalled Carbon Nanotubes for Drug Delivery Applications. Adv.Sci.Lett. 2012, 17, 70-75. https://doi.org/10.1166/asl.2012.4251].
4. In materials sections authors need to add Cell section, properly added all cell used in this manuscript along with cell culture protocols.
Answer: Detailed protocol as desired by the reviewer has been incorporated in the materials and method section of the manuscript as follows:
Cell culture and treatments: The Human breast cancer cell line (MCF7) was procured from the American Type Cutler Collection (Manassas, VA, USA). The cells were maintained in Dulbecco's Modified Eagle Medium (DMEM) with phenol red supplement with 10% Fetal Bovine Serum (FBS), with Penicillin (100 units/mL), Streptomycin (100 μg/mL), and of Amphotericin B (250 ng/mL), Gibco® (New York, USA). The cells were grown at 37°C in 50 cm2 tissue culture flasks in 5% CO2 humidified incubator. The cells were seeded into 96-well cell culture plates in DMEM.
5. Why morphological changes observed in breast cancer?
Answer: The morphological changes were observed in MCF7 cells by Sunitinib as it is a tyrosine kinase inhibitor and has anti-proliferative effect on breast cancer cell lines (MCF-7). The Sunitinib treatment cause concentration dependent cell growth suppression due to apoptosis as evident by Caspase-3, p53 and Caspase-9 levels in MCF7. These results were consistent and similar previous reports [PMID: 28870911: Anticancer Res. 2017 Sep;37(9):4899-4909. doi: 10.21873/anticanres.11899.; PMID: 23288144; Arch Toxicol. 2013 May;87(5):847-56. doi: 10.1007/s00204-012-0996-y. Epub 2013 Jan 4.].
6. Authors also need to perform in vitro cytotoxicity study of hybrid Nps on Normal cell to confirm safety.
Answer: We appreciate the suggestion of the reviewer. Because, normal cell lines are not available in our facility and it was difficult to get human breast tissues to make normal cell lines. In future studies of hybrid NPs of Sunitinib, we will perform a detailed mechanistic study on nude mice as well as normal cell lines.
7. for size, PDI, and ZP results authors need to draw proper graphs on graphpad. Kindly remove Table 2. for these results authors also need to insert proper results obtained from the zeta sizer and DLS.
Answer: As suggested, Table 2 has been replaced with Figure 2 in revised manuscript.
8. For drug release study how the sample were analyzed by using 250 nm? Which instrument is used, HPLC or UV?
Answer: UV/Vis spectroscopy (Jasco) method.
9. Methodology for In vitro drug release study need to re write again, kindly add time interval b/w two samples taken and after taken sample how to maintain sink condition?
Answer: Methodology of release study has been revised.
10. Discussion for SEM result need improvement.
Answer: Improved.
11. In fig 7 there is no morphological diffrence observed, kindly performed it again.
Answer: As suggested, a new figure 7 has been added after repetition of experiment.
12. Its better draw all graphs by using Graphpad and apply all the T test, Excel graphs are not attractive.
Answer: As suggested, Graph pad generated (Figure 2) has been added in revised manuscript. data were analyzed statistically using “one way ANOVA” followed by “Tukey’s multiple comparison test using SPSS 16 software (p < 0.01 was considered significant
13. Why not author checked this Hybrid Nps loaded with anticancer drug on Animal? Molecular studies such as H&E staining is missing. Kindly perform it.
Answer: We appreciate the suggestion of the reviewer. Because, animals models and H & E staining facilities were not available. In future studies of hybrid NPs of Sunitinib, we will perform a detailed anticancer study on animal.
14. Discussion part need to revised to and compare your results with already published literature to support your hypothesis.
Answer: As suggested, the discussion section has been improved with suitable references.
15. Please revisit the entire manuscript for minor grammar and typo issues.
Answer: Done.
Reviewer 3 Report
After careful evaluation, I recommend publication of this paper in Polymers after revisions. The problems should be concerned are listed below:
- there are several grammar and typo errors throughout the manuscript, for instance, line 168.
- all the tables should be replacced with three-line table.
- there are too many figures in the manuscript, acutally some figures can be integrated into one.
- the interior structure, for instance, the cross-section structure, should be demonstrated.
- in figure.7, the cell morphology can not be clearly observed, please use SEM image instead.
Author Response
Dear Sir/madam,
We appreciate the reviewer for your precious time in reviewing our paper and providing valuable comments. It was your valuable and insightful comments that led to possible improvements in the current version. The authors have carefully considered the comments and tried our best to address every one of them.
We hope the manuscript after careful revisions meet your high standards. The authors welcome further constructive comments if any. Below we provide the point-by-point responses. All modifications in the manuscript have been done in track change.
After careful evaluation, I recommend publication of this paper in Polymers after revisions. The problems should be concerned are listed below:
- there are several grammar and typo errors throughout the manuscript, for instance, line 168.
Answer: Grammar and typo errors throughout the manuscript has been corrected.
2. all the tables should be replacced with three-line table.
Answer: As suggested, all figure has been formatted in three line.
3. there are too many figures in the manuscript, actually some figures can be integrated into one.
the interior structure, for instance, the cross-section structure, should be demonstrated.
Answer: We appreciate the suggestion of the reviewer. These facilities were not available. In future these studies will be performed.
4. in figure.7, the cell morphology can not be clearly observed, please use SEM image instead.
Answer: As suggested, a new figure 7 has been added after repetition of experiment.
Round 2
Reviewer 2 Report
Accepted in present form
Reviewer 3 Report
The quality of this paper has been greatly improved, thus I recommend publication of this paper in Polymers.